# PMA-qPCA: Accelerating the market release of high-quality *Bradyrhizobium diazoefficiens* inoculant

Mariana Cap [1,2]*, Camila Frydman[2], Antonella Galiñanes[3], Camila Aranguiz[3], Irina Faraco[3], Luisina Andriolo[3], Viviana Parreño[4], Marina Mozgovoj [1,2]*

**1** INTA, Instituto Nacional de Tecnología Agropecuaria, Instituto Tecnológico de Alimentos, Hurlingham, Argentina, **2** ICYTESAS, Instituto de Ciencia y Tecnología de Sistemas Alimentarios Sustentables, INTA-CONICET, Hurlingham, Argentina, **3** Rizobacter S.A., Pergamino, Argentina, **4** INTA-IVIT-INCUINTA, Instituto de Virología e Innovaciones Tecnológicas, Hurlingham, Argentina

\* cap.mariana@inta.gob.ar (MC); mozgovoj.marina@inta.gob.ar (MM)

## Abstract

Traditional culture-based quantification of *Bradyrhizobium diazoefficiens* in inoculants presents significant limitations due to its labor-intensive and time-consuming nature. To address this limitation, we aimed to validate a propidium monoazide quantitative PCR (PMA-qPCR) assay as a rapid and reliable alternative for estimating *Bradyrhizobium diazoefficiens* counts in commercial inoculants. Key experiments optimized PMA concentration (50, 75 and 100 µM) to selectively inhibit DNA amplification from non-viable cells without interfering with viable cell signal. Assay´s efficiency, limit of detection and quantification, intra-assay repeatability and inter-assay reproducibility were determined. The assay demonstrated high efficiency (90–105%), a limit of detection (LOD) of 3.14 log CFU/mL, and a dynamic range from 8.74 to 3.14 log CFU/mL. Robust intra-assay repeatability (SD < 0.3) and inter-assay reproducibility (CV < 10%) were confirmed. The method successfully distinguished quarter-strength and 10-fold serial dilutions of viable bacteria, even in the presence of non-viable cells. Final validation against standard plate counting showed a strong linear correlation with an $R^2$ of 0.82. Crucially, this PMA-qPCR assay reduced processing time from 120 hours to just 5 hours, offering a significant improvement in turnaround time while maintaining strong agreement with the reference method. This study marks the first application of PMA-qPCR for *Bradyrhizobium diazoefficiens* quantification in inoculants, highlighting its potential as a high-throughput tool to enhance efficiency and precision for industrial batch-to-batch quality control.

## 1. Introduction

The increasing concern over climate change and the need for more sustainable agricultural practices have positioned *Bradyrhizobium* as a valuable tool for legume inoculation. These bacteria establish symbiotic relationships with plants by forming root

---

---

**Data availability statement:** All relevant data are within the paper and its Supporting Information files.

**Funding:** This study was funded by private funds from Rizobacter S.A. and public funds from the research project 2019-PE-E7-I120 of the National Institute of Agricultural Technology.

**Competing interests:** The authors have declared that no competing interests exist.

or stem nodules, where they fix atmospheric nitrogen and convert it into ammonia, a form readily available for plant uptake. However, the growing demand for *Bradyrhizobium* inoculants has revealed a major bottleneck in the production process: the lengthy and labor-intensive viability certification [1,2]. Currently, the viability assessment relies on plate counting, a method that requires several days of incubation, delaying the product´s release to the market. This creates significant pressure on inoculant manufacturers to optimize their processes and adopt more efficient certification methods. To overcome the challenge of ensuring both rapid product delivery and compliance with strict quality standards, the use of PMA-qPCR was evaluated.

Propidium Monoazide (PMA) is a membrane-impermeant dye that selectively penetrates non-viable cells with compromised membranes. Upon photoactivation, PMA´s highly reactive nitrene radical forms covalent bonds with double-stranded DNA, primarily through insertion and cross-linking reactions, thereby preventing its amplification during PCR. This covalent modification, rather than simple intercalation, is key to its irreversible binding and efficacy in distinguishing between viable and non-viable cells in molecular assays by excluding DNA from dead cells [3,4]. This approach has been successfully applied by other researchers for various matrices and bacterial species. For instance, Bouju-Albert et al. (2021) [5], validated PMA-qPCR for quantifying viable *Brochotrix thermosphacta* in cold-smoked salmon. Shi et al. (2022) [6], used PMA-qPCR to selectively quantify viable lactic acid bacteria in fermented milk. In this regard, we have previously demonstrated the effectiveness of PMA-qPCR in detecting and quantifying viable Shiga toxin-producing *Escherichia coli* in beef burgers after treatment with high hydrostatic pressure [7,8].

This study aimed to evaluate the effectiveness of PMA-qPCR for quantifying viable *Bradyrhizobium diazoefficiens* in commercial inoculants. The aim of this study was to optimize the qPCR assay for bacterial quantification, standardize the PMA-qPCR protocol for viable bacterial quantification, and finally validate the methodology using the final product.

## 2. Materials and methods

### 2.1. Bacterial strain, growth conditions and traditional microbiological quantification

*Bradyrhizobium diazoefficiens (strain SEMIA 5080),* obtained from the Rizobacter S.A. culture collection, was employed for assay standardization in this study, which was conducted from March 2024 to March 2025. For quantification using the plate count method, bacterial cultures were serially diluted in peptone water 0.1%, and spread onto modified Mannitol Yeast Agar with Congo Red (MYA+CR). Plates were incubated at $29.5 \pm 1°C$ for 4–5 days or until colonies reached a diameter of 1.5 to 3 mm. Results were expressed as the $Log_{10}$ of colony-forming units (CFU) per mL.

### 2.2. DNA extraction

Genomic DNA was extracted using TIANamp Genomic DNA Kit (Tiangen, China) following the manufacture instructions for Gram negative bacterial DNA. Samples consisted of bacterial cultures or commercial inoculants (final products), as appropriate.

## 2.3. qPCR

Primers and TaqMan probe were designed based on conserved regions identified through sequence alignment of *Bradyrhizobium diazoefficiens* strains SEMIA 5080 (accession NZ_ADOU02000008.1) using the Nucleotide BLAST tool. Primers and probe sequences were as follows: primer FW 5´GGAAGCAAGCGACAAGTCT 3´; primer REV 5´TTCGT-GCTCGACAATCTCAC 3´ and Probe 5´FAM-TCGTTAACCATCCGTCAACCAGCA-MGB-EDQ 3´. Real-time PCR was performed using the StepOnePlus™ System (Thermo Fisher Scientific, USA). The reactions volume was 12.5 µl volume, containing 5 µM of 2X iTaq Universal Probes Supermix (Bio-Rad, USA); 0.5 µM of each primer, 0.25 µM of the probe, and 4.7 µl of DNA template. The thermal cycling parameters were 95°C for 5 min followed by 40 cycles at 95°C for 15 s and 60°C for 1 s.

## 2.4. Optimization of qPCR for *Bradyrhizobium diazoefficiens* quantification

Initially, a total of six independent assays were conducted using ten serial 5-fold dilutions of *Bradyrhizobium diazoefficiens* DNA, spanning a range from 8.74 to 2.45 log CFU/mL. Each assay was performed in triplicate, resulting in 18 replicates per concentration and a total of 180 measurements. The qPCR limit of detection (LOD), limit of quantification (LOQ), repeatability, dynamic range, and amplification efficiency were estimated following the MIQE guidelines [9,10]. The LOD was specifically calculated by analyzing the results of independent 6 runs of the standard curve as described in Fig 1a. To assess the agreement between qPCR-based estimates and traditional plate counts, a simple linear regression model was applied. The value of 1 CFU/mL was included as the baseline of the dose–response curve and was considered non-detectable. Samples with Ct values below 40 were considered positive, based on the upper limit of the prediction interval established for the relationship between Ct value and bacterial plate counts (CFU/mL).

The data were analyzed using two mathematical models for discrete data. Both methods are referred to as mechanistic, as they assume that the bacterial particles in the sample follow a Poisson distribution around the mean dose. While the exponential model assumes a constant binomial probability that a viable bacterial unit survives in the sample matrix and its genome is detectable by the technique, the beta-Poisson model assumes that this probability varies according to a beta distribution [11]. Finally, the specificity of the qPCR assay was evaluated by testing 20 phylogenetically related strains.

## 2.5. Optimization of PMA-qPCR for *Bradyrhizobium diazoefficiens* quantification

**2.5.1 PMA treatment.** PMA (PMAxx, Biotium Inc., USA) was diluted in DEPC water (diethylpyrocarbonate-treated, deionized, filtered water, UltraPure, Invitrogen, USA) to a working concentration of 5 mM and stored at −20°C. PMA treatment was performed according to the manufacturer´s instructions. To each 400 µL sample, 100 µl of PMA Enhancer (Biotium Inc., USA) was added, followed by a volume of 5 mM PMA to reach a final concentration of 50, 75 or 100 µM. Subsequently, the samples were incubated with mechanical shaking at room temperature for 30 min wrapped in aluminum foil to avoid exposure to light. Then, they were exposed to blue LED light (PhAST Blue, Geniul, Spain) for 15 min to achieve photoactivation of the dye. Finally, the excess of PMA was removed by centrifugation at 8000 xg for 10 min (Centrifuge 5417C, Eppendorf, Germany) and the supernatant was discarded.

**2.5.2. Experiment 1 - Minimal concentration of PMA to inhibit non-viable cell signal.** To determine the minimal effective concentration of PMA for inhibiting the signal from non-viable cells, 10-fold dilutions of *Bradyrhizobium diazoefficiens* (SEMIA 5080) in exponential phase were heat-inactivated (95°C for 10 min). The resulting dilutions (5–9 log CFU/mL) were treated with 50, 75 and 100 µM of PMA. DNA was extracted and analyzed by qPCR in triplicate.

**2.5.3. Experiment 2 - Assessment of potential interference of PMA on the amplification of different concentrations of viable cells.** Two aliquots of three serial dilutions of an exponential phase bacterial suspension (6, 7 and 8 log CFU/mL) were prepared. One aliquot was treated with PMA while the other served as control. Data were analyzed

## a- Standartization curve

| Bacterial plate count (CFU/ml) | Log$_{10}$ (CFU/ml) | Positive | Negative | Total |
|---|---|---|---|---|
| 5.44E+08 | 8.74 | 18 | 0 | 18 |
| 1.09E+08 | 8.04 | 18 | 0 | 18 |
| 2.18E+07 | 7.34 | 18 | 0 | 18 |
| 4.35E+06 | 6.64 | 18 | 0 | 18 |
| 8.70E+05 | 5.94 | 18 | 0 | 18 |
| 1.74E+05 | 5.24 | 18 | 0 | 18 |
| 3.48E+04 | 4.54 | 18 | 0 | 18 |
| 6.96E+03 | 3.84 | 14 | 4 | 18 |
| 1.39E+03 | 3.14 | 12 | 6 | 18 |
| 2.79E+02 | 2.45 | 4 | 14 | 18 |

## b- Dynamic range

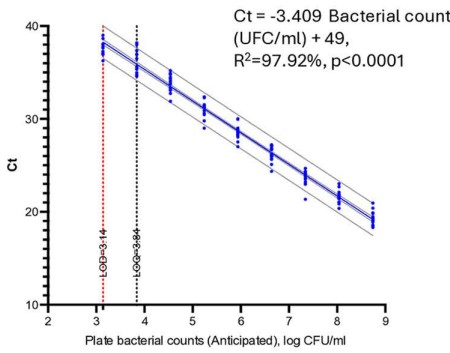

Ct = -3.409 Bacterial count (UFC/ml) + 49, $R^2$=97.92%, p<0.0001

## c- Comparison of beta Poisson and Exponential model fit for LOD estimation

| MODEL | MINIMIZED DEVIANCE | DIFFERENCE BETWEEN DEVIANCES | CHI-SQUARED CRITICAL | CHI-SQRD P-value | CONCLUSION |
|---|---|---|---|---|---|
| Exponential | 9.2125 | 6.0615 | 3.8415 | 0.0138 | beta Poisson model is the BEST fitting model |
| Beta Poisson | 3.151 | | | | |

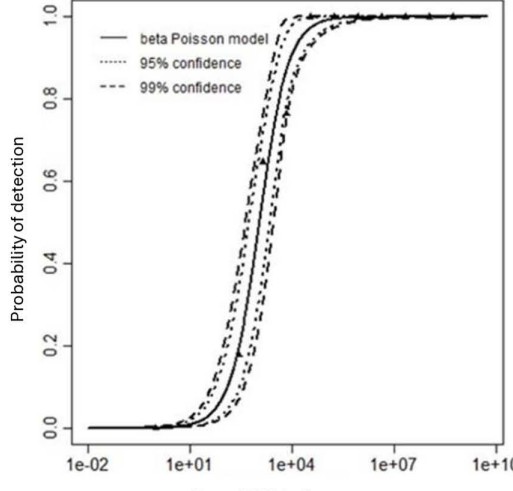

d- Probability of detection as a function of bacterial counts (CFU/ml). The solid curve represents the Beta-Poisson model, and the dashed curves indicate the CI 95% and 99% confidence intervals.

## e- LOD and LOQ

| Minimized Deviance | α | N50 = LDPr50 | LOD95 | LOQ |
|---|---|---|---|---|
| 3.151 | 1.086 | 1126.6= 10e3.05 CFU/ml | 1390.83= 10e3.14 CFU/ml | 6960= 10e3.84 CFU/ml |

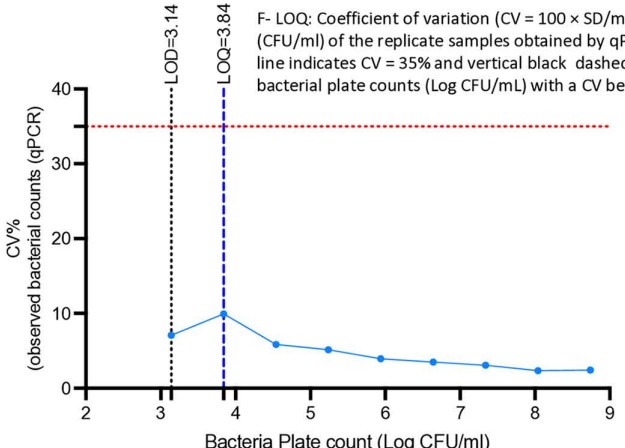

F- LOQ: Coefficient of variation (CV = 100 × SD/mean) of bacterial counts (CFU/ml) of the replicate samples obtained by qPCR. Horizontal red dashed line indicates CV = 35% and vertical black dashed line indicates the lowest bacterial plate counts (Log CFU/mL) with a CV below 35%. (Forootan, 2017)

**Fig 1. Optimization of qPCR for Bradyrhizobium diazoefficiens quantification.** (a) Standard curve: Detection rate of 10 serial 5-fold dilutions of bacterial DNA, each tested in six independent qPCR assays with three replicates per dilution. (b) Detection rate as a function of the bacterial dose (CFU/mL). The solid line represents the Beta-Poisson model, and the dashed lines show the 95% and 99% confidence intervals. Parameters of the Beta-Poisson model (alfa and N50) and LOD estimated from the fitted curve are indicated. (c) Comparison of model fit for LOD estimation using Beta-Poisson and Exponential models. (d) Dynamic range of qPCR and consensus standard curve with 95% confidence and prediction intervals. The limits of detection (LOD) and quantification (LOQ) of the assay are indicated. (e) Linear regression between the viable bacterial estimated by PMA-qPCR and bacterial plate count. (f) LOQ estimation based on the curve of the coefficient of variation (CV% = 100 × SD/mean) as a function of bacterial plate counts (log CFU/mL). The horizontal red dashed line indicates CV = 35%, and the vertical black dashed line marks the LOD and the LOQ defined as the lowest bacterial concentration (log CFU/mL) at which CV falls below 35%.

using two-way ANOVA, considering PMA treatment (with or without) as a fixed effect and bacterial concentration as another fixed effect. Post-hoc comparisons were performed using Tukey´s test. Statistical significance was set at $p < 0.05$.

**2.5.4. Experiment 3 - Evaluation of the PMA-qPCR's accuracy in quantifying viable bacteria in the presence of a high concentration of non-viable cells.** A *Bradyrhizobium diazoefficiens* culture in exponential phase of 8.64 log CFU/mL was heat-inactivated (95°C for 10 min) and fractionated into 4 tubes, each containing 900 µl. Subsequently, 100 µl of viable bacteria was added and serial 10-fold dilutions were performed using the tubes containing non-viable bacteria. Each tube was treated with PMA as described in 2.5.1. The aim was to evaluate four concentrations of live bacteria (7.64, 6.64, 5.64 and 4.64 log CFU/mL) in the presence of a high concentration of dead bacteria (8.64 log CFU/mL). Data were analyzed using one-way ANOVA, considering bacterial concentration as fixed effect. Post-hoc comparisons were performed using Tukey´s test. Statistical significance was set at $p < 0.05$.

**2.5.5. Experiment 4 -Determination of PMA-qPCR effectiveness in distinguishing viable from non-viable bacteria within the same concentration range.** A *Bradyrhizobium diazoefficiens* culture in exponential phase of 8.64 log CFU/mL was heat-inactivated (95°C for 10 min) and fractionated into 4 tubes, each containing 500 µl. Subsequently, 500 µl of viable bacteria was added and serial 2-fold dilutions were performed using the tubes containing non-viable bacteria. Each tube was treated with PMA as described in 2.5.1. The aim was to evaluate four concentrations of live bacteria (8.43, 8.13, 7.83 and 7.53 log CFU/mL) within the same order of magnitude in the presence of a high concentration of dead bacteria (8.64 log CFU/mL). Data were analyzed using one-way ANOVA, considering bacterial concentration as fixed effect. Post-hoc comparisons were performed using Tukey´s test. Statistical significance was set at $p < 0.05$.

**2.5.6. Experiment 5 - Validation of the methodology using batches of the final product at different storage times.** A total of 3 assays were conducted; each assay included 3 samples of the commercial product. Each sample had different storage time (0, 12, 24 months post-production) and therefore different concentrations of viable and non-viable bacteria. Since the expected concentrations exceeded the dynamic range of the qPCR, we worked with 1:100 dilutions. Samples were analyzed using both, PMA-qPCR and conventional plate count to estimate the number of live bacteria. A linear regression analysis was performed to evaluate the agreement between both methods, with the aim of validating PMA-qPCR as a predictive tool for viable bacterial quantification, potentially replacing plate counting in-process quality control of the product. To evaluate the effect of PMA treatment (including samples without PMA as controls), data were analyzed using one-way ANOVA, considering PMA treatment as fixed effect. Post-hoc comparisons were performed using Tukey´s test. Statistical significance was set at $p < 0.05$.

## 3. Results

### 3.1 qPCR performance parameters

The efficiency values for all assays were within the acceptable range (90–105%), indicating high-quality qPCR data. The LOD, defined as the lowest concentration of DNA that can be reliably detected, was determined to be 3.14 log CFU/mL using the beta-Poisson model. The LOQ, defined as the lowest detectable concentration within the linear range of the standard curve, was found to be 3.84 log CFU/mL. The dynamic range, defined as the range of

concentration over which the standard curve is linear, was determined to be between 8.74 and 3.14 Log CFU/mL (Fig 1). The specificity assay showed that none of the analyzed strains generated a qPCR signal. The intra-assay repeatability, expressed as the standard deviation of the replicates at each point of the standard curve, was less than 0.3 in all cases. Similarly, the inter-assay reproducibility, expressed as the coefficient of variation between runs, was less than 10% (Supplementary material).

### 3.2. Experiment 1 - Minimal concentration of PMA to inhibit non-viable cell signal

For bacterial concentrations ranging from 5 to 8 log CFU/mL), no detectable qPCR signal was observed across all three tested PMA concentrations (50, 75 and 100 µM), indicating complete inhibition of DNA amplification from non-viable bacteria at these concentrations. Although all tested concentrations proved effective, 50 µM was selected for all subsequent assays to optimize resource utilization. However, a detectable fluorescence signal was observed when treating heat-inactivated bacteria at a concentration of 9 log CFU/mL with 50, 75 and 100 µM PMA (S1 Table). This indicates that 8 log CFU/mL represents the maximum allowable concentration of non-viable bacteria in the system for complete signal inhibition under these conditions.

### 3.3. Experiment 2 - Assessment of potential interference of PMA on the amplification of different concentrations of viable cells

Analysis revealed no statistically significant differences in viable bacteria counts between PMA-treated and untreated samples for any of the concentration tested (6–8 log CFU/mL). These results indicate that PMA did not interfere with the signal from viable bacteria (Fig 2a).

### 3.4. Experiment 3 - Evaluation of the PMA-qPCR's accuracy in quantifying viable bacteria in the presence of a high concentration of non-viable cells

PMA-qPCR significantly distinguished among 10-fold serial dilutions of viable bacteria (7.64, 6.64, 5.64 and 4.64 log CFU/mL counts), even when diluted in a fixed concentration of non-viable bacteria ($p < 0.001$; one-way ANOVA) (Fig 2b).

### Experiment 4 – Determination of PMA-qPCR effectiveness in distinguishing viable from non-viable bacteria within the same concentration range

PMA-qPCR results were inconsistent when discriminating between 2-fold serial dilutions (8.43, 8.13, 7.83 and 7.53 log CFU/mL). Nevertheless, the technique showed good sensitivity in discriminating 4-fold serial dilutions (one-way ANOVA, Tukey) (Fig 2c).

### 3.5. Experiment 5 - Validation of the methodology using batches of the final product at different storage times

Following the linear regression analysis, it was observed that $R^2$ was 0.82, indicating that the model explains 82% of the variability of the dependent variable. The regression line, the confidence bands and the 95% prediction bands are illustrated in Fig 3a. As a control, samples without PMA were included. Unlike the PMA-treated samples, they showed no statistically significant differences across the analyzed time points (Fig 3b).

## 4. Discussion

Quantification of *Bradyrhizobium* in inoculants currently relies heavily on cultured-based assays, which are time-consuming and labor-intensive. However, accurate quantification is essential to ensure the quality and efficacy of these inoculants. Therefore, the development and validation of rapid, reliable and high-throughput methods, such as the qPCR, are critical for practical agricultural applications.

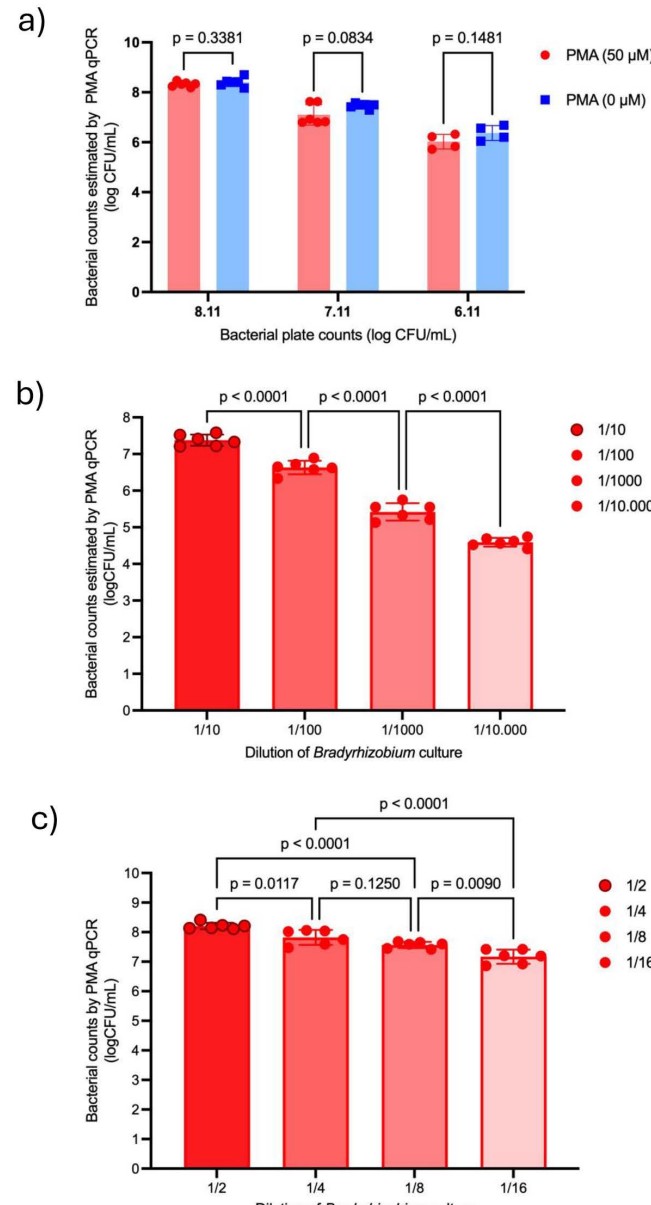

**Fig 2. Optimization of PMA-qPCR for quantifying *Bradyrhizobium diazoefficiens*.** (a) Experiment 2, Assessment of potential interference of PMA on the amplification of different concentrations of viable cells, two-way ANOVA, Tukey. b) Experiment 3: Evaluation of the PMA-qPCR accuracy in quantifying viable bacteria in the presence of a high concentration of non-viable cells (serial10-fold dilutions of the bacterial culture; 1/10 =7.64 log CFU/mL viables in 8,64 log CFU/mL non-viable bacteria). c) Experiment 4: Assessment of PMA-qPCR performance in distinguishing viable from non-viable cells across a serial 2-fold concentration range (dilution of the culture 1/2=8.43 log CFU/mL viables in 8,64 log CFU/mL non-viable bacteria). One-way ANOVA-Tukey.

In this study, we validated a PMA-qPCR assay for the quantification of viable *Bradyrhizobium* as an alternative to conventional culture methods. The performance of the qPCR was demonstrated through the following parameters: an efficiency of approximately 95%, with intra-assay repeatability indicated by a standard deviation of 0.3, and an inter-assay

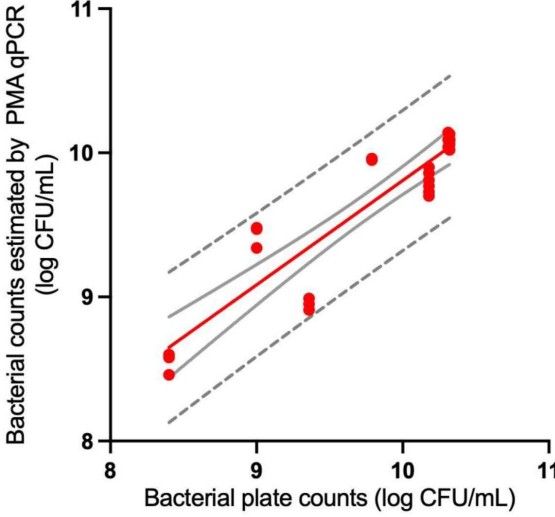

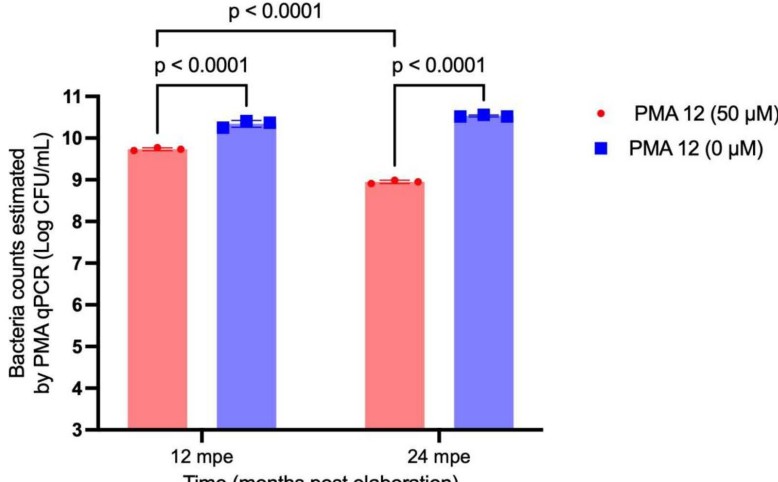

**Fig 3. Validation of the methodology using batches of the final product at three different storage times (0-, 12- and 24-months post-manufacturing).** (a) Lineal regression between the viable bacteria estimated by PMA-qPCR and the bacterial plate count. (b) Comparison of mean bacterial counts between different storage times (12- and 24-months post elaboration), with and without PMA treatment, repeated measure Anova-Tukey.

reproducibility with a coefficient of variation less than 10%. These results confirm the reliability and robustness of the qPCR technique for estimating *Bradyrhizobium diazoefficiens* counts.

The LOD and LOQ were also estimated, reflecting the sensitivity of the technique. Although they were slightly higher than those reported for other applications, they are well suited for the intended purpose of this study [12]. A more precise determination of LOD and LOQ could be achieved by performing serial 2-fold dilutions from a 1:1000 dilution onward, following CEFAS recommendations [13]. However, this was beyond the scope of this work, as the primary goal was to estimate the proportion of viable and non-viable cells for batch-to-batch quality control of a product containing viable cell counts in the order of 10 log CFU/mL to 8 log CFU/mL, far beyond the conservative LOD selected for the assay.

Optimizing the concentration of PMA is a crucial step that depends heavily on the intended use of the technique. In our case, samples at time zero were expected to have a viable cell concentration of 10 log CFU/mL. After two years of storage, samples were expected to have a viable cell concentration of 8 log CFU/mL and a non-viable cell concentration of 9.99 log CFU/mL. Thus, the PMA had to effectively inhibit the signal from a high number of non-viable cells in the presence of a high number of viable cells. Since these concentrations exceeded the dynamic range of the qPCR, we worked with 1:100 dilutions, resulting in samples with 6 log CFU/mL of viable cells and 7.99 log CFU/mL of non-viable cells.

To determine the minimum PMA concentration required for complete inhibition of signal from non-viable bacteria, we first tested non-viable cells alone across a range of concentration of PMA concentrations (4–9 log CFU/mL, Experiment 1). Once the minimum PMA concentration was selected, it was evaluated against different concentrations of viable bacteria to ensure it did not interfere with their accurate quantification (Experiment 2). Finally, various concentrations of viable cells were assessed in the presence of a high concentration of non-viable cells (Experiment 3). These assays demonstrated that PMA-qPCR could effectively distinguish among 10-fold serial dilutions (7.64, 6.64, 5.64 and 4.64 log CFU/mL counts), even in the presence of high levels of non-viable cells (8.64 log CFU/mL). To validate the technique´s discrimination threshold a series of twofold dilutions were conducted (Experiment 4). However, while it was unable to resolve between half-strength dilutions, it successfully differentiated quarter-strength dilutions. This result was expected, as the technique validation had already demonstrated a standard deviation of 0.3 log CFU/mL. Given that the difference between two samples diluted to half-strength is also 0.3 log CFU/mL, the findings are consistent with previous assessments.

As a final assessment (Experiment 5), validation of the method was carried out against the standard plate counting technique, achieving a strong linear correlation with an $R^2$ of 0.82. In other words, PMA-qPCR method was able to determine the number of viable cells in just 5h, compared to the 120 hours required by conventional plate counting, while confirming its strong agreement with the reference method. This nearly 96% reduction in turnaround time enables faster product release, minimizes inventory holding, and improves market responsiveness. This efficiency translates primarily into reduced labor costs, as PMA-qPCR drastically decreases manual hands-on time per sample, allowing for greater automation and optimized workforce utilization at scale. While qPCR reagents might seem more expensive per test than basic culture media, the overall cost per validated result often becomes more favorable with molecular techniques due to less incubator space, fewer consumables, and the prevention of costly product spoilage or re-processing from delayed results. In this regard, PMA-qPCR offers superior scalability, as high-throughput qPCR systems can simultaneously process hundreds of samples, a capacity impractical with manual plating, making it an ideal solution for large scale production and rapid quality control.

Regarding the correlation of 0.82, even though it is considered satisfactory, it is important to identify potential causes for the remaining 18% of variability not explained by the model. In this highly controlled system, the variance is more likely attributable to subtle biological variations within inoculants physiological states that can still introduce slight measurements noise in a sensitive quantitative assay. Furthermore, as methodology scales up in industrial use, thousands of data points will be collected allowing for a more comprehensive understanding of this variability. This increase data will contribute to further process refinements, leading to an improved $R^2$ value, thereby demonstrating an even more robust and reliable correlation with the standard plate count technique in practical industrial applications.

To the best of our knowledge, this study is the first to apply PMA-qPCR specifically for quantifying *Bradyrhizobium diazoefficiens* in agricultural inoculants. Pedrolo et al. (2024) [12]evaluated this technique for *Herbaspirillum seropedicae*, an endophytic diazotroph, in pure culture and maize roots, and demonstrate that PMA-qPCR is a powerful approach for quantifying viable and viable but nonculturable cells (VBNC) in inoculants and maize roots. In contrast to the aforementioned study, our results showed no significant differences between plate counts and concentrations estimated from the standard curve. While *Bradyrhizobium diazoefficiens* can indeed enter a VBNC state under various stresses, our experiments were conducted using a liquid inoculant matrix specifically formulated to maintain bacterial viability, including the presence of cryoprotectants and appropriate substrates. Cryoprotectants, commonly used in bacterial formulations and

lyophilization, are known to stabilize cell membranes and cellular components, thereby preventing damage that can lead to loss of culturability and transition to VBNC state. Previous studies have demonstrated that the presence of protective agents such as trehalose and glycerol can significantly reduce cell damage and inhibit the entry intro de VBNC state under different environmental stresses, including desiccation and temperature fluctuations, by stabilizing bacterial membranes [14,15]. While direct studies specifically linking cryoprotectants to VBNC prevention in *Bradyrhizobium* are limited, the general mechanism applies across bacteria, supporting our observation.

Another challenge faced by inoculant producers is that if the culture medium becomes contaminated or if inoculants are applied to non-sterile materials, traditional quantification techniques become difficult. This is due to the variable presence of Gram-positive bacteria and fungi, which can interfere with counting accuracy. Recent studies have proposed improvements to culture media to make them more selective [16,17]. However, none have achieved complete elimination of the accompanying microbiota. In this context, the PMA-qPCR technique stands out for its high specificity.

## 5. Conclusions

This study successfully validated a PMA-qPCR assay for rapid and reliable quantification of viable *Bradyrhizobium diazoefficiens* in inoculants, offering a strong alternative to traditional plate counting. The method demonstrated high efficiency, reproducibility, and selectivity, effectively distinguishing viable cells even in the presence of high non-viable cell concentrations. Validation against plate counting confirmed an 82% correlation, reducing processing time from 120 hours to just 5 hours. This substantial shift represents a significant economic advantage, streamline operations, reduced overheads, and make faster, more informed decisions about product quality and release. These findings highlight PMA-qPCR as a valuable tool for microbial viability assessment in industrial applications.

## Supporting information

**S1 Table. Optimization of qPCR for *Bradyrhizobium diazoefficiens* quantification.**
(XLSX)

**S2 Table. Minimal PMA concentration to inhibit non-viable cell signal.**
(XLSX)

**S3 Table. Experiment 2, Assessment of potential interference of PMA on the amplification of different concentrations of viable cells.**
(XLSX)

**S4 Table. Experiment 3: Evaluation of the PMA-qPCR accuracy in quantifying viable bacteria in the presence of a high concentration of non-viable cells.**
(XLSX)

**S5 Table. Experiment 4: Assessment of PMA-qPCR performance in distinguishing viable from non-viable cells across a serial 2-fold concentration range.**
(XLSX)

**S6 Table. Lineal regression between the viable bacteria estimated by PMA-qPCR and the bacterial plate count.**
(XLSX)

**S7 Table. Comparison of mean bacterial counts between different storage times (12- and 24-months post elaboration), with and without PMA treatment.**
(XLSX)

## Acknowledgments

We appreciate the participation of Solange Galeano for technical assistance.

## Author contributions

**Conceptualization:** Mariana Cap, Antonella Galiñanes, Camila Aranguiz, Irina Faraco, Luisina Andriolo, Marina Mozgovoj.

**Data curation:** Mariana Cap, Camila Frydman, Antonella Galiñanes, Camila Aranguiz, Irina Faraco, Luisina Andriolo, Marina Mozgovoj.

**Formal analysis:** Mariana Cap, Camila Frydman, Marina Mozgovoj.

**Funding acquisition:** Marina Mozgovoj.

**Investigation:** Mariana Cap, Camila Aranguiz, Irina Faraco, Luisina Andriolo, Marina Mozgovoj.

**Methodology:** Mariana Cap, Camila Frydman, Antonella Galiñanes, Viviana Parreño, Marina Mozgovoj.

**Project administration:** Mariana Cap, Luisina Andriolo, Marina Mozgovoj.

**Supervision:** Mariana Cap, Marina Mozgovoj.

**Validation:** Mariana Cap, Viviana Parreño, Marina Mozgovoj.

**Visualization:** Mariana Cap, Marina Mozgovoj.

**Writing – original draft:** Mariana Cap, Viviana Parreño, Marina Mozgovoj.

**Writing – review & editing:** Mariana Cap, Luisina Andriolo, Viviana Parreño, Marina Mozgovoj.

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
