## [Decision Letter · Decision Letter 0]

19 Jul 2025

Dear Dr. Cap,

Thank you for submitting your manuscript to PLOS ONE. After careful consideration, we feel that it has merit but does not fully meet PLOS ONE’s publication criteria as it currently stands. Therefore, we invite you to submit a revised version of the manuscript that addresses the points raised during the review process.

We look forward to receiving your revised manuscript.

Kind regards,

Grzegorz Woźniakowski, Full professor, PhD, ScD

Academic Editor

PLOS ONE

Journal requirements

This study was funded by private funds from Rizobacter S.A. and public funds from the research project 2019-PE-E7-I120 of the National Institute of Agricultural Technology.

This study was funded by private funds from Rizobacter S.A. and public funds from the research project 2019-PE-E7-I120 of the National Institute of Agricultural Technology. We appreciate the participation of Solange Galeano for technical assistance.

This study was funded by private funds from Rizobacter S.A. and public funds from the research project 2019-PE-E7-I120 of the National Institute of Agricultural Technology.

5. Please amend your list of authors on the manuscript to ensure that each author is linked to an affiliation. Authors’ affiliations should reflect the institution where the work was done (if authors moved subsequently, you can also list the new affiliation stating “current affiliation:….” as necessary).

7. Please include a separate caption for each figure in your manuscript.

8. Please remove all personal information, ensure that the data shared are in accordance with participant consent, and re-upload a fully anonymized data set.

Reviewers' comments:

Reviewer's Responses to Questions

**Comments to the Author**

1. Is the manuscript technically sound, and do the data support the conclusions?

Reviewer #1: Yes

Reviewer #2: Yes

Reviewer #3: Yes

2. Has the statistical analysis been performed appropriately and rigorously?

Reviewer #1: I Don't Know

Reviewer #2: Yes

Reviewer #3: Yes

3. Have the authors made all data underlying the findings in their manuscript fully available?

Reviewer #1: Yes

Reviewer #2: Yes

Reviewer #3: Yes

4. Is the manuscript presented in an intelligible fashion and written in standard English?

Reviewer #1: Yes

Reviewer #2: No

Reviewer #3: Yes

Reviewer #1: Greetings

Dear authers, this manuscript evaluated the effectiveness of PMA-qPCR for quantifying

viable Bradyrhizobium diazoefficiens in commercial inoculants.

Kindly note: In methods you did not write the date or period of current study for

example from May 2024 to December 2024.This is important scientifically.

Kind regards.

Reviewer #2: Abstract:

Line 17 = Please clarify what is the meaning of "Approximately 95% efficiency", efficiency of what? qPCR amplification?

Line 21= Discrimination threshold assessment” is vague — consider clarifying what this means (e.g., dilution sensitivity? detection limit?

The abstract would benefit from aligning with a standard of scientific format: Background, Objective, Methods, Results, Conclusion/impact.

Keywords could be expanded to better reflect the focus of the research.

Introduction:

Line 43-48 = The PMA explanation should be streamlined.

Line 52-54 = the last sentence shifts to “we demonstrated…” this implies the authors of the current paper have already published this prior work. If not, please revise to clarify the authorship.

Line 55-58 = Consider stronger emphasize for objective statement (ex: This study evaluates the use of PMA-qPCR to quantify viable Bradyrhizobium diazoefficiens in commercial inoculants, including assay optimization, PMA protocol refinement, and validation on finished products).

Methods:

Line 63 = Please indicate the type of diluent used for serial dilutions and the dilution range. Please specify what is the meaning of modified Mannitol Yeast Agar with Congo Red?

Line 68 = Clarify the kit name, and add a brief info what kind of sample was used?

Line 69 = manufacture instructions  manufacturer's instructions

Line 84 & Line 86 shows the redundancy of LOD, please consider to merge this statement for clarity.

Line 91 = unclear phrasing "Samples were considered positive when Ct values were below 40" should be "Samples with Ct values below 40 were considered positive...."

Line 103 = The “xx” should be replaced with the actual catalog number or information (e.g., PMAxx or PMA-Lite), or removed if unknown.

Line 106 = "Briefly, 100 µl of PMA Enhancer...” Briefly” is often overused, it can be removed or replaced with more scientific phrasing (µl should consistently be written as “µL” (capital L), ml should be mL according to standard SI unit formatting).

Line 107 = ".............. to the desired final concentration" is unclear, should be revised "...........to reach to final concentration of 50, 75, or 100 µM."

Line 117 = analyze should be analyzed (use past tense throughout the method).

Line 155 = lineal .....should be linear

Line 158 = "For the latter...." is vague, consider for merging for clarity.

Line 190-192 = The sentence would benefit from a brief contextual explanation (e.g., what mixture was tested, and whether viable cells were diluted in a constant background of non-viable cells). Consider defining the experimental condition: were the non-viable cells added at a fixed concentration? p should be italicized for publication style.

Line 215 = “intra-assay repeatability and standard deviation of 0.3” should be “...with intra-assay repeatability indicated by a standard deviation of 0.3”. “an intra assay reproducibility” → should be “inter-assay reproducibility” based on context (between runs), and needs a hyphen in “inter-assay”.

Line 243 = failed to distinguish should be more formal such as was unable to resolve

Line 244 = "half strength ....... and quarter strength ....... " should be written as "half-strength and quarter-strength"

Line 250 = "was able to determine the number of viable cells in only just 5 2 h" should be written as "...... in just 5.2 hours....." (make sure the decimal is accurate and consistent).

Line 251 = " ....while maintaining an 82%....." is redundant with earlier sentence and could be omitted or restated more precisely.

Line 265 = " some researchers ......" avoid vague phrases, change to be more precise

Be consistent with species names (italicize Herbaspirillum seropedicae, Bradyrhizobium diazoefficiens).

Reviewer #3: Major Comments

1. Methodological Clarity

o PMA Optimization (Section 2.5.1): The manuscript states that 50 µM PMA was selected for subsequent assays but does not explicitly justify why this concentration was optimal (e.g., statistical comparison of inhibition efficacy across 50, 75, and 100 µM). Include a quantitative analysis (e.g., % signal inhibition at each concentration) to support this choice.

o Heat Inactivation Controls: Clarify whether heat inactivation (95°C for 10 min) was validated to ensure complete cell death without DNA fragmentation, which could affect PMA binding efficiency. A viability stain (e.g., SYTOX Green) could corroborate this.

2. Statistical Analysis

o Figure 2c (Experiment 4): The inability to distinguish 2-fold dilutions contrasts with the stated SD of 0.3 log CFU/mL. Reconcile this discrepancy by discussing technical limitations (e.g., pipetting error, PMA heterogeneity) or proposing a higher dilution factor for finer discrimination.

o Regression Analysis (Figure 3a): The R² of 0.82 suggests unexplained variability. Discuss potential sources (e.g., matrix effects, PMA penetration variability) and how they might be mitigated in industrial applications.

3. Comparative Advantages

o Time/Cost Savings: Emphasize the economic impact of reducing processing time from 120 h to 5 h. Include a brief cost comparison (e.g., reagents, labor) between PMA-qPCR and plate counting to highlight scalability.

o VBNC State: The dismissal of VBNC cells (lines 259–261) lacks experimental evidence. Cite supporting data or reference prior studies showing cryoprotectants prevent VBNC formation in Bradyrhizobium.

4. Industrial Applicability

o Sample Throughput: Describe how the method scales for high-throughput batch testing (e.g., number of samples processed per run, automation potential).

o Matrix Variability: Address whether the method was tested across different inoculant formulations (e.g., peat-based vs. liquid carriers) to ensure broad applicability.

Minor Comments

1. Abstract

o Replace "82% correlation" with "R² = 0.82" for statistical precision.

2. Introduction

o Line 52: Clarify how PMA’s nitrene radical binds DNA (covalent vs. intercalation) to aid reader understanding.

3. Results

o Figure 1: Label axes in all panels (e.g., "Log CFU/mL" for bacterial counts).

o Line 165: Specify which model (beta-Poisson or exponential) yielded the LOD of 3.14 log CFU/mL.

4. Discussion

o Line 226: Expand on why LOD/LOQ values are acceptable despite being higher than other applications (e.g., inoculants typically have high cell densities).

5. References

o Ensure all citations (e.g., Pedrolo et al., 2024) are in the bibliography.

**Do you want your identity to be public for this peer review?** For information about this choice, including consent withdrawal, please see our Privacy Policy

Reviewer #1: No

Reviewer #2: No

Reviewer #3: **Yes: ** Maryam Zakavi

---

## [Author Response · Author response to Decision Letter 1]

25 Aug 2025

Response to editor and reviewers

Dear editor and reviewers,

We appreciate all the comments, corrections and suggestions and the time employed that significantly improve our manuscript. Please, find below the response to each raised point.

Sincerely,

Mariana Cap, in the name of all authors

Reviewer #1: Greetings

Dear authers, this manuscript evaluated the effectiveness of PMA-qPCR for quantifying

viable Bradyrhizobium diazoefficiens in commercial inoculants.

Kindly note: In methods you did not write the date or period of current study for

example from May 2024 to December 2024.This is important scientifically.

Kind regards.

R: The detailed period was included in the manuscript as following: March 2024 to March 2025 (Lines 70-71).

Reviewer #2:

Abstract:

Line 17 = Please clarify what is the meaning of "Approximately 95% efficiency", efficiency of what? qPCR amplification?

R: approximately was removed from the sentence as suggested. Abstract was modified for clarity as suggested (Lines 16-33).

Line 21= Discrimination threshold assessment” is vague — consider clarifying what this means (e.g., dilution sensitivity? detection limit?

R: The sentences was replaced by: The method successfully distinguished quarter-strength and 10-fold serial dilutions of viable bacteria, even in the presence of non-viable cells. (Lines 26-27).

The abstract would benefit from aligning with a standard of scientific format: Background, Objective, Methods, Results, Conclusion/impact.

R: the abstract was modified as suggested (Lines 16-33).

Keywords could be expanded to better reflect the focus of the research.

R: PMA-qPCR and cell viability were added (Line 35).

Introduction:

Line 43-48 = The PMA explanation should be streamlined.

R: PMA explanation was modified to be streamlined as suggested (Lines 49-54).

Line 52-54 = the last sentence shifts to “we demonstrated…” this implies the authors of the current paper have already published this prior work. If not, please revise to clarify the authorship.

R: the sentence refers to previous published work of our research group (https://doi.org/10.1111/jfpp.15338;
https://doi.org/10.1111/1750-3841.16179;
https://doi.org/10.1111/1750-3841.70109). To clarify, the sentence was corrected as follows: …”we have previously demonstrated…” (Lines 58-59).

Line 55-58 = Consider stronger emphasize for objective statement (ex: This study evaluates the use of PMA-qPCR to quantify viable Bradyrhizobium diazoefficiens in commercial inoculants, including assay optimization, PMA protocol refinement, and validation on finished products).

R: the objective was changed as suggested: The aim of this study was to optimize the qPCR assay for bacterial quantification, standardize the PMA-qPCR protocol for viable bacterial quantification, and finally validate the methodology using the final product (Lines 63-65).

Methods:

Line 63 = Please indicate the type of diluent used for serial dilutions and the dilution range. Please specify what is the meaning of modified Mannitol Yeast Agar with Congo Red?

R: dilutions were performed in peptone water 0.1%. Mannitol Yeast Agar with Congo Red is the culture medium specific for Bradyrhizobium. This information was included in the manuscript (Line 72).

Line 68 = Clarify the kit name, and add a brief info what kind of sample was used?

R: The kit name is Tianamp Bacteria DNA kit (https://en.tiangen.com/content/details_43_4220.html). Genomic DNA from bacteria was extracted from bacterial cultures or commercial inoculants. This was included in the manuscript (Lines 77-79)

Line 69 = manufacture instructions  manufacturer's instructions

R: corrected as suggested

Line 84 & Line 86 shows the redundancy of LOD, please consider to merge this statement for clarity.

R: The sentence was modified for clarity as suggested: “The qPCR limit of detection (LOD), limit of quantification (LOQ), repeatability, dynamic range, and amplification efficiency were estimated following the MIQE guidelines (9, 10). The LOD was specifically calculated by analyzing the results of independent 6 runs of the standard curve (Figure 1a)” (Lines 95-98).

Line 91 = unclear phrasing "Samples were considered positive when Ct values were below 40" should be "Samples with Ct values below 40 were considered positive...."

R: corrected as suggested (Line 101).

Line 103 = The “xx” should be replaced with the actual catalog number or information (e.g., PMAxx or PMA-Lite), or removed if unknown.

R: PMAxx was corrected as suggested. (Line 114 )

Line 106 = "Briefly, 100 µl of PMA Enhancer...” Briefly” is often overused, it can be removed or replaced with more scientific phrasing (µl should consistently be written as “µL” (capital L), ml should be mL according to standard SI unit formatting).

R: corrected as suggested. The sentence was replaced by: To each 400 µL sample, 100 µl of PMA Enhancer (Biotium Inc., USA) was added, followed by a volume of 5 mM PMA to reach a final concentration of 50, 75 or 100 µM (Lines 116-118). Furthermore, volumes were corrected according to standard SI unit formatting.

Line 107 = ".............. to the desired final concentration" is unclear, should be revised "...........to reach to final concentration of 50, 75, or 100 µM."

R: corrected as suggested: To each 400 �L sample, 100 µl of PMA Enhancer (Biotium Inc., USA) was added, followed by a volume of 5 mM PMA to reach a final concentration of 50, 75 or 100 µM) (Lines 116-118).

Line 117 = analyze should be analyzed (use past tense throughout the method).

R: corrected (Line 129).

Line 155 = lineal .....should be linear

R: Corrected (Line 167).

Line 158 = "For the latter...." is vague, consider for merging for clarity.

R: the paragraph was improved for clarity as it follows: “A linear regression analysis was performed to evaluate the agreement between both methods, with the aim of validating PMA-qPCR as a predictive tool for viable bacterial quantification, potentially replacing plate counting in-process quality control of the product. To evaluate the effect of PMA treatment (including samples without PMA as controls), data were analyzed using one-way ANOVA, considering PMA treatment as fixed effect. Post-hoc comparisons were performed using Tukey´s test. Statistical significance was set at p < 0.05” (Lines 167-172).

Line 190-192 = The sentence would benefit from a brief contextual explanation (e.g., what mixture was tested, and whether viable cells were diluted in a constant background of non-viable cells). Consider defining the experimental condition: were the non-viable cells added at a fixed concentration? p should be italicized for publication style.

R: The sentence was modified according to reviewer´s suggestion: PMA-qPCR significantly distinguished among 10-fold serial dilutions of viable bacteria (7.64, 6.64, 5.64 and 4.64 log CFU/ml counts), even when diluted in a fixed concentration of non-viable bacteria (p<0.001; one-way ANOVA) (Figure 2b) (Lines 206-208).

Line 215 = “intra-assay repeatability and standard deviation of 0.3” should be “...with intra-assay repeatability indicated by a standard deviation of 0.3”. “an intra assay reproducibility” → should be “inter-assay reproducibility” based on context (between runs), and needs a hyphen in “inter-assay”.

R: corrected as suggested: with intra-assay repeatability indicated by a standard deviation of 0.3, and an inter-assay reproducibility (Lines 268-269).

Line 243 = failed to distinguish should be more formal such as was unable to resolve

R: corrected as suggested (Lines 297).

Line 244 = "half strength ....... and quarter strength ....... " should be written as "half-strength and quarter-strength"

R: corrected as suggested (Line 298).

Line 250 = "was able to determine the number of viable cells in only just 5 2 h" should be written as "...... in just 5.2 hours....." (make sure the decimal is accurate and consistent).

R: The word only was removed as suggested. It is just 5 h, not 5.2 h (Line 304).

Line 251 = " ....while maintaining an 82%....." is redundant with earlier sentence and could be omitted or restated more precisely.

R: The sentence was modified for improvement as suggested: …while confirming its strong agreement with the reference method (Lines 305-306).

Line 265 = " some researchers ......" avoid vague phrases, change to be more precise

Be consistent with species names (italicize Herbaspirillum seropedicae, Bradyrhizobium diazoefficiens).

R: The sentences was modified as suggested: Recent studies have proposed improvements to culture media to make them more selective (Line 304).

Reviewer #3: Major Comments

1. Methodological Clarity

o PMA Optimization (Section 2.5.1): The manuscript states that 50 µM PMA was selected for subsequent assays but does not explicitly justify why this concentration was optimal (e.g., statistical comparison of inhibition efficacy across 50, 75, and 100 µM). Include a quantitative analysis (e.g., % signal inhibition at each concentration) to support this choice.

R: To better explain the choice of 50 �M PMA, the paragraph was corrected as follows: For bacterial concentrations ranging from 5 to 8 log CFU/mL), no detectable qPCR signal was observed across all three tested PMA concentrations (50, 75 and 100 �M), indicating complete inhibition of DNA amplification from non-viable bacteria at these concentrations. Although all tested concentrations proved effective, 50 µM was selected for all subsequent assays to optimize resource utilization. However, a detectable fluorescence signal was observed when treating heat-inactivated bacteria at a concentration of 9 log CFU/mL with 50, 75 and 100 µM PMA (S1 Table). This indicates that 8 log CFU/mL represents the maximum allowable concentration of non-viable bacteria in the system for complete signal inhibition under these conditions (Lines 189-196).

Furthermore, to determine the minimal effective concentration of PMA for inhibiting the signal from non-viable cells, 10-folf dilutions of Bradyrhizobium diazoefficiens (SEMIA 5080) in exponential phase were heat-inactivated (95°C for 10 min). This information was included in M&M section (Lines 126-128). The S1 Table was included in Supporting Material.

o Heat Inactivation Controls: Clarify whether heat inactivation (95°C for 10 min) was validated to ensure complete cell death without DNA fragmentation, which could affect PMA binding efficiency. A viability stain (e.g., SYTOX Green) could corroborate this.

R: Our results, presented in Section 2.5.1 (PMA Optimization) and specifically in Supporting Material (Table S1), demonstrated a complete inhibition of qPCR signal from heat-inactivated bacteria across a broad range of concentrations (4-8 log CFU/mL), using 50, 75 and 100 µM PMA. This indirectly confirmed that any potential DNA fragmentation did not significantly impede PMA`s ability to bind effectively. Furthermore, previous studies of our research group demonstrated that similar heating conditions effectively permeabilized bacterial membrane for viability assays without causing significant DNA degradation that would render it unsuitable for PMA-qPCR (https://doi.org/10.1111/1750-3841.70109; DOI: 10.1111/1750-3841.16179; https://doi.org/10.1111/jfpp.15338), thus reinforcing the suitability and effectiveness of the inactivation method for the specific purpose of generating non-viable cells for PMA-qPCR standardization in this study.

2. Statistical Analysis

o Figure 2c (Experiment 4): The inability to distinguish 2-fold dilutions contrasts with the stated SD of 0.3 log CFU/mL. Reconcile this discrepancy by discussing technical limitations (e.g., pipetting error, PMA heterogeneity) or proposing a higher dilution factor for finer discrimination.

R: The inability to distinguish 2-fold dilutions in Experiment 4, despite a stated SD of 0.3 log CFU/mL, is not a contradiction but rather a demonstration of the assay´s practical limits when the difference being measured is equal to its inherent variability. The difference between two samples diluted to half-strength (2-fold dilution) is 0.3 log CFU/mL, while the SD of our validated method is 0.3. In this regard, the method can not discriminate among 2-fold dilutions.

o Regression Analysis (Figure 3a): The R² of 0.82 suggests unexplained variability. Discuss potential sources (e.g., matrix effects, PMA penetration variability) and how they might be mitigated in industrial applications.

R: This is a good observation. Regarding the correlation of 0.82 even though it is considered satisfactory, it is important to identify potential causes for the remaining 18% of variability not explained by the model. In this highly controlled system, the variance is more likely attributable to subtle biological variations within inoculants´ physiological states that can still introduce slight measurements noise in a sensitive quantitative assay. Furthermore, as methodology scales up in industrial use, thousands of data points will be collected allowing for a more comprehensive understanding of this variability. This increase data will contribute to further process refinements, leading to an improved R2 value, thereby demonstrating an even more robust and reliable correlation with the standard plate count technique in practical industrial applications. This discussion was incorporated in the Discussion section (Lines 317-325).

3. Comparative Advantages

o Time/Cost Savings: Emphasize the economic impact of reducing processing time from 120 h to 5 h. Include a brief cost comparison (e.g., reagents, labor) between PMA-qPCR and plate counting to highlight scalability.

R: The standardized PMA-qPCR method delivers transformative time and cost savings compared to traditional plate counting, significantly boosting industrial economic efficiency by reducing viability assessment form 120 h to just 5h. The following paragraph was included in the discussion section: This nearly 96% reduction in turnaround time enables faster product release, minimizes inventory holding, and improves market responsiveness. This efficiency translates primarily into reduced labor costs, as PMA-qPCR drastically decreases manual hands-on time per sample, allowing for greater automation and optimized workforce utilization at scale. While qPCR reagents might seem more expensive per test than basic culture media, the overall cost per validated result often becomes more favorable with molecular techniques due to less incubator space, fewer consumables, and the prevention of costly product spoilage or re-processing from delayed results. In this regard, PMA-qPCR offers superior scalability, as high-throughput qPCR systems can simultaneously process hundreds of samples, a capacity impractical with manual plating, making it an ideal solution for large scale production and rapid quality control (Lines 305-316). The following sentence was included in the conclusion: This substantial shift represents a significant economic advantage, streamline operations, reduced overheads, and make faster, more informed decisions about product quality and release. (Lines 358-360).

o VBNC State: The dismissal of VBNC cells (lines 259–261) lacks experimental evidence. Cite supporting data or reference prior studies showing cryoprotectants prevent VBNC formation in Bradyrhizobium.

R: This is a good observation. The following paragraph was included for clarity: While Bradyrhizobium diazoefficiens can indeed enter a VBNC state under various stresses, our experiments were conducted using a liquid inoculant matrix specifically formulated to maintain bacterial viability, including the presence of cryoprotectants and appropriate substrates. Cryoprotectants, commonly used in bacterial formulations and lyophilization, are known to stabilize cell membranes and cellular components, thereby preventing damage that can lead to loss of culturability and transition to VBNC state. Previous studies have demonstrated that the presence of protective agents such as trehalose and glycerol can significantly reduce cell damage and inhibit the entry intro de VBNC state under different environmental stresses, including desiccation and temperature fluctuations, by stabilizing bacterial membranes (https://doi.org/10.1016/s0011-2240(03)00046-4 ; https://doi.or

---

## [Decision Letter · Decision Letter 1]

2 Sep 2025

PMA-qPCR: ACCELERATING THE MARKET RELEASE OF HIGH-QUALITY BRADYRHIZOBIUM DIAZOEFFICIENS INOCULANTS

PONE-D-25-27271R1

Dear Dr. Cap,

We’re pleased to inform you that your manuscript has been judged scientifically suitable for publication and will be formally accepted for publication once it meets all outstanding technical requirements.

Kind regards,

Mohammad H. Ghazimoradi

Academic Editor

PLOS ONE

Additional Editor Comments (optional):

Reviewer #1:

Reviewer #3:

Reviewers' comments:

Reviewer's Responses to Questions

**Comments to the Author**

Reviewer #1: All comments have been addressed

Reviewer #3: All comments have been addressed

2. Is the manuscript technically sound, and do the data support the conclusions?

Reviewer #1: Yes

Reviewer #3: Yes

3. Has the statistical analysis been performed appropriately and rigorously?

Reviewer #1: Yes

Reviewer #3: Yes

4. Have the authors made all data underlying the findings in their manuscript fully available?

Reviewer #1: Yes

Reviewer #3: Yes

5. Is the manuscript presented in an intelligible fashion and written in standard English?

Reviewer #1: Yes

Reviewer #3: Yes

Reviewer #1: Greetings,

The revisions have significantly improved the manuscript. It is now scientifically stronger and suitable for publication.

Kind regards.

Reviewer #3: Thank you for submitting your paper to PLOS One. Now all the comments have been addressed, and the manuscript could be accepted.

**Do you want your identity to be public for this peer review?** For information about this choice, including consent withdrawal, please see our Privacy Policy

Reviewer #1: No

Reviewer #3: **Yes: ** Maryam Zakavi

---

## [Editor Report · Acceptance letter]

PONE-D-25-27271R1

PLOS ONE

Dear Dr. Cap,

I'm pleased to inform you that your manuscript has been deemed suitable for publication in PLOS ONE. Congratulations! Your manuscript is now being handed over to our production team.

Kind regards,

on behalf of

Dr. Mohammad H. Ghazimoradi

Academic Editor

PLOS ONE